# PFC: A Novel Perceptual Features-Based Framework for Time Series Classification

**DOI:** 10.3390/e23081059

**Published:** 2021-08-17

**Authors:** Shaocong Wu, Xiaolong Wang, Mengxia Liang, Dingming Wu

**Affiliations:** 1College of Computer Science and Technology, Harbin Institute of Technology, Shenzhen 518055, China; wushaocong2013@gmail.com (S.W.); hitsz_wudingming@outlook.com (D.W.); 2College of Computer Science and Technology, Harbin Institute of Technology, Harbin 150001, China; liangmengxia@hotmail.com

**Keywords:** time series classification, perceptual features, decision trees, ensemble learning

## Abstract

Time series classification (TSC) is a significant problem in data mining with several applications in different domains. Mining different distinguishing features is the primary method. One promising method is algorithms based on the morphological structure of time series, which are interpretable and accurate. However, existing structural feature-based algorithms, such as time series forest (TSF) and shapelet traverse, all features through many random combinations, which means that a lot of training time and computing resources are required to filter meaningless features, important distinguishing information will be ignored. To overcome this problem, in this paper, we propose a perceptual features-based framework for TSC. We are inspired by how humans observe time series and realize that there are usually only a few essential points that need to be remembered for a time series. Although the complex time series has a lot of details, a small number of data points is enough to describe the shape of the entire sample. First, we use the improved perceptually important points (PIPs) to extract key points and use them as the basis for time series segmentation to obtain a combination of interval-level and point-level features. Secondly, we propose a framework to explore the effects of perceptual structural features combined with decision trees (DT), random forests (RF), and gradient boosting decision trees (GBDT) on TSC. The experimental results on the UCR datasets show that our work has achieved leading accuracy, which is instructive for follow-up research.

## 1. Introduction

In the information age, massive amounts of data have been generated over time. These data are closely related to many studies. In mathematics, a time series is a series of data points indexed in time order. Most commonly, a time series [1] is a sequence taken at successive equally spaced points in time. Time series contains information on time dimension and data dimension, and it exists in many fields such as economy, life science, military science, space science, geology and meteorology, and industrial automation. Time series classification [2,3,4] is an essential task that has attracted widespread attention. Normally, time series classification refers to assign time series patterns to a specific category, for example, judge whether it will rain or not through a series of temperature data [5] or determine whether the patient has Parkinson’s disease through a period of physiological data [6,7]. Dau et al. [8] proposed UCR Time Series Classification Archive (UCR) for this task, including 128 datasets from different fields such as ECG, Sensor, and Image. In order to understand TSC more intuitively, Figure 1 shows some representative datasets in UCR. These datasets almost cover the existing TSC tasks, show the morphological structure of various time series, and lay the foundation for researchers to explore general classification methods. In order to solve this problem, many methods have been proposed, which can be divided into five categories according to their different cores: dictionary-based, distance-based, interval-based, shapelet-based, and kernel-based.

The dictionary-based method refers to the idea of natural language processing. Researchers believe that a time series is a special sentence composed of discrete words or words. How to segment and map the time series into characters is the first issue that needs to be considered. There are three main time series symbolization methods: Piecewise Aggregate Approximation (PAA) [9,10], Symbolic Aggregate approXimation (SAX) [11,12], and Symbolic Fourier Approximation (SFA) [13]. Subsequently, the Bag-of-SFA Symbols (BOSS) method based on the bag-of-words model was proposed [14]. This method records high-frequency symbol features and uses them to distinguish different types of time series samples. Matthew et al. [15] and James et al. [16] further proposed Contract BOSS (cBOSS) and Spatial BOSS (S-BOSS). In addition, Word Extraction for Time Series Classification (WEASEL) [17] is also a typical dictionary-based method composed of a supervised symbolic time series representation for discriminative word generation and the Bag of Patterns (BOP) [18] model for building a discriminative feature vector.

Many TSC methods focus on the distance between time series. Generally, a time series can be regarded as a point in a multi-dimensional space, and the dimension of this multi-dimensional space depends on the length of the time series. Different types of time series will have different aggregations. At this time, distance is an effective way to distinguish. K-Nearest Neighbors (KNN) and the Elastic Ensemble (EE) [19] are two commonly used methods. Ben et al. [20] proposed Proximity Forest to model a decision tree forest that uses distance measures to partition data. It should be noted that since most distance calculations use the form of “one to one”, samples of equal length are necessary. For unequal length sequences, dynamic time warping (DTW) [21,22,23] is a robust calculation method, which can avoid differences in length and shape. Combining KNN and DTW is a way to take advantage of both at the same time [24,25].

In reality, different types of time series may have precisely the same statistical characteristics such as mean, variance, standard deviation, and so on [26]. In order to avoid this problem, the interval-based method focuses on local features rather than overall features. Deng et al. [27] proposed a Time Series Forest (TSF) model that converts time series into statistical features of sub-sequences and uses random forest for classification. Cabello et al. [28] further constructed Supervised Time Series Forest (STSF), an ensemble of decision trees built on intervals selected through a supervised process. Random Interval Spectral Ensemble (RISE) is a popular variant of time series forest [29]. RISE differs from time series forest in two ways. First, it uses a single time series interval per tree. Second, it is trained using spectral features extracted from the series instead of summary statistics. Since RISE relies on frequency information extracted from the time series, it can be defined as a frequency-based classifier.

The shapelet-based method draws inspiration from pattern recognition. Shapelets are defined in [30,31] as “subsequences that are in some sense maximally representative of a class”. Informally, if we assume a binary classification setting, a shapelet is discriminant if it is present in most classes and absent from the series of the other class. However, any subsequence may be distinguishable, and the length of the subsequence is arbitrary, which means that all samples and their subsequences need to be checked through a sliding window, and the search space for shapelets is enormous. In response to this problem, Ji et al. [32,33] proposed a fast shapelets selection algorithm.

Building on the recent success of convolutional neural networks for time series classification, Dempster et al. [34] realize that simple linear classifiers using random convolutional kernels achieve state-of-the-art accuracy with a fraction of the computational expense of existing methods. Therefore, they proposed ROCKET, a kernel-based time series classification method. This is a new direction for TSC, which can both reduce computational complexity and improve accuracy.

By analyzing the five classification methods, we realized that the existing algorithms are essentially trying to find efficient distinguishing features by learning all the original information of the sample, which leads to high computational complexity and resource consumption. In fact, for human beings, it does not require all the information to distinguish time series. On the contrary, we only pay attention to a few critical data points, which are enough to describe the approximate outline of time series samples and present a significant distribution. This paper proposes a classification framework based on perceptual features, which can extract support points of morphological structure from the original time series and further obtain interval-level and point-level features for classifiers such as decision trees. The contributions of our work are described below.

An improved algorithm called globally restricted matching perceptually important points (GRM-PIPs) is proposed, which avoids the redundancy caused by sequential matching in traditional important point extraction.How many data points are necessary to describe complete information? We conducted in-depth research on this question and verified our opinions through mathematical proofs and experiments.The data points extracted by GRM-PIPs can divide the time series into sub-sequences similar to shapelets. We used statistical features such as mean, standard deviation, slope, skewness, and kurtosis to enhance discrimination further.Most classifiers learn the information of the original time series, which is not suitable for perceptual features. Therefore, we matched a suitable classifier and proposed a complete perceptual features-based framework.

The remainder of this paper is organized as follows. In Section 2, related work about PIPs, decision trees, random forests, and gradient boosting decision trees are presented. Section 3 describes the details about PFC, including GRM-PIPs, perceptual feature extraction, and classifiers adaptation. Section 4 presents the experimental setup and performance of the approach we proposed, as well as a comparison of the experiments and performances. A discussion about the differences in experimental results is also given in Section 4. Finally, the conclusions and directions for future research are given in Section 5.

## 2. Related Work

### 2.1. Perceptually Important Points

For time series, avoiding point-to-point local comparison is the key to reducing computational complexity. In time series pattern mining, unique, and frequently occurring patterns can be abstractly represented by several critical points. It is precisely through these points with important visual impacts that humans remember specific time series patterns [35]. The definition of perceptually important points was first introduced in reference [36]. The PIPs algorithm can retain the key turning points in the time series, and its ability to capture the critical points in the time series has been verified in the time series segmentation and pattern recognition [37,38,39].

Interestingly, PIPs have been widely used in the research of stock time series. Fu et al. [40] used PIPs as a new time series segmentation method to extract the uptrend and downtrend patterns. Mojtaba et al. [41] regard PIPs as a dimensionality reduction method similar to PCA and combine it with support vector regression to predict the trend of the stock market. The turning point in the stock time series indicates a substantial change in the market, and PIPs are sensitive to these dividing points, which is also the advantage of PIPs.

In general, we would define any time series as $T=\left\{t_{1},t_{2},\ldots ,t_{n}\right\}\;\;n\in Z^{+}$. This is a classic one-dimensional definition, which treats a time series as a string of data arranged chronologically. However, a one-dimensional data sequence is considered to have no morphological structure and cannot be displayed on a two-dimensional plane. Therefore, we need to upgrade the traditional one-dimensional definition to two-dimensional to explain the calculation process of PIPs. By introducing data in the time dimension, the two-dimensional definition of a time series is $T=\left\{\left(x_{1},y_{1}\right),\left(x_{2},y_{2}\right),\ldots ,\left(x_{n},y_{n}\right)\right\}\;\;n\in Z^{+}$, where $x_{n}$ represents the current data point in the $n^{th}$ position in the entire time series, and $y_{n}$ corresponds to amplitude. PIPs uses a concise idea to extract important points in the morphological structure of time series. The process is shown below.

**Definition** **1.*****Perceptually Important Points***.
*Given a time series sample $T=\left\{\left(x_{1},y_{1}\right),\left(x_{2},y_{2}\right),\ldots ,\left(x_{n},y_{n}\right)\right\},n>2,\;n\in Z^{+}$, an empty list $L_{p}$ is set to save the extracted perceptually important points. In general, when extracting m important points, the following steps should be followed.*

***Step 1***
*: Put the first point $\left(x_{1},y_{1}\right)$ and the last point $\left(x_{n},y_{n}\right)$ in T as initial two PIPs into $L_{p}$.*

***Step 2***
*: Check each point in T and calculate the distance between them and $\left(x_{1},y_{1}\right)$ and $\left(x_{n},y_{n}\right)$. Choose the point with the largest distance as the third PIP and save it in $L_{p}$.*

***Step 3***
*: The fourth PIP is the point that maximizes its distance to its adjacent PIPs (which are either the first and the third, or the third and the second PIP). It is also necessary to save the fourth PIP into the $L_{p}$.*

***Step 4***
*: For each new PIP, use the same method as the fourth PIP, repeat Step 3 until the length of $L_{p}$ is equal to m.*


For PIPs, there are three distance measures including the euclidean distance (ED), the perpendicular distance (PD), and the vertical distance (VD). The calculation formula of the vertical distance $VD\left(P_{c}\right)$ between $P_{c}\left(x_{c},y_{c}\right)$ and the line $P_{a}P_{b}$ is shown in Formula (1) and Figure 2. We use VD to show the calculation process of PIPs through a simple example.
(1)$$VD\left(P_{c}\right)=\left|y_{c}-y_{d}\right|=\left|y_{c}-\left(\frac{\left|x_{b}-x_{a}\right|\cdot \left|x_{d}-x_{a}\right|}{\left|y_{b}-y_{a}\right|}+y_{a}\right)\right|$$

We define the extraction of *m* PIPs from the sample *T* as $PIPs\left(T,\;\;m\right)$. For example, if the one-dimensional representation of *T* is $T=\left\{0,4,2.5,3,1,5,6,1,2,1,0\right\}$, its corresponding two-dimensional representation as below. The process of finding six PIPs from *T* is shown in Figure 3.


$T=\left\{\left(0,0\right),\left(1,4\right),\left(2,2.5\right),\left(3,3\right),\left(4,1\right),\left(5,5\right),\left(6,6\right),\left(7,1\right),\left(8,2\right),\left(9,1\right),\left(10,0\right)\right\}$


It should be noted that when there are two points with the same maximum vertical distance, the first calculated point is usually set as the new PIP.

### 2.2. Decision Tree and Ensemble Methods

A Decision Tree (DT) is a decision support tool that uses a tree-like model of decisions and their possible consequences, including chance event outcomes, resource costs, and utility. It is one way to display an algorithm that only contains conditional control statements. DT is a non-parameter supervised learning method used for classification and regression. Its purpose is to create a model to learn simple decision rules from data features to predict the value of a target variable [42]. DT are commonly used in operations research, specifically in decision analysis, to help identify a strategy most likely to reach a goal, but are also a popular tool in machine learning [43].

DT is a predictive model in machine learning, which represents a mapping relationship between object attributes and object values. Each node in the tree represents an object, and each bifurcation path represents a possible attribute value, and each leaf node corresponds to the value of the object represented by the path from the root node to the leaf node. The decision tree has only a single output. If you want to have a complex output, you can build an independent decision tree to handle different outputs [44]. Simultaneously, DT is a frequently used technique in data mining, which can be used to analyze data, and it can also be used to make predictions [45].

The applications of decision tree on the TSC task mainly has three directions, pattern recognition, shapelet transformation and features selection. Pierre [4] believed that many time-series classification problems can be solved by detecting and combining local properties or patterns in time series and he proposed a technique based on DT to find patterns which are useful for classification. Qiu et al. [46] forecast shanghai composite index based on fuzzy time series and improved C-fuzzy decision trees. Willian et al. [47] explored shapelet transformation for time series classification in decision trees and develop strategies to improve the representation quality of the shapelet transformation. In essence, the DT uses the “if-then-else” rule to learn the data, and the deeper the rule is applied, the better the data fitting will be.

There are too many research results and knowledge about DT, and we would not repeat them specifically. The following is a simple example to introduce DT. We assume a scenario that includes three factors: season, wind, and time. In this scenario, record the data of whether someone is doing morning exercises, as shown in Table 1. This scenario is a typical classification task, and the decision tree constructed based on Table 1 is shown in Figure 4.

The ensemble methods is a high-level application, and the decision tree is regarded as a basic/weak estimator. The goal of the ensemble methods is to combine the predictions of multiple basic estimators to achieve better generalization or robustness than a single estimator. The ensemble methods generally fall into three categories:

**Bagging Method.** This method usually considers the homogeneous weak estimators, learns these weak estimators independently and in parallel, and combines them according to some deterministic average process [48]. In general, the combined estimator is better than the single estimator because its variance is reduced. Random forest (RF) [49] is a typical Bagging method, it can build a large number of decision trees to filter features to get the best decision rule set.**Boosting Method.** The core of this method is also a combination of homogeneous weak estimators. It sequentially learns these weak estimators in a highly adaptive method (each basic estimator attempts to reduce the bias of the combined estimator), and combines them according to a certain deterministic strategy. The current popular boosting methods include AdaBoost and Gradient Tree Boosting. Freund and Schapire proposed the former in 1999 [50]. Its core idea is to train a series of weak estimators by repeatedly modifying the weights of the data [51]. On the other hand, Gradient Tree Boosting [52] is a generalization of the lifting algorithm for any differentiable loss function. It can be used for classification and regression and applied to various fields, including web search ranking and ecological environment [53,54].**Stacking Method.** Different from the previous methods, the stacking method uses heterogeneous estimators, learns them in parallel, and combines them by training a “meta mode” to output a final result according to different predictions [55].

## 3. Perceptual Features-Based Framework

This section will introduce the perceptual features-based framework (PFC) in detail, divided into three parts: time series preprocessing with GRM-PIPs, feature extraction, and classifier. These parts have a precise sequence in our framework.

### 3.1. Time Series Preprocessing with GRM-PIPs

The purpose of this part is to traverse the time series and extract a certain number of PIPs. Based on the traditional PIPs algorithm, we determined the global optimal selection strategy and proposed a restrictive selection method. The relevant definition is as follows.

**Definition** **2.*****Globally Restricted Matching Perceptually Important Points***.
*Given a time series sample $T=\left\{\left(x_{1},y_{1}\right),\left(x_{2},y_{2}\right),\ldots ,\left(x_{n},y_{n}\right)\right\},n>2,\;n\in Z^{+}$, an empty list $L_{p}$ is set to save the extracted perceptually important points, the interval between adjacent PIPs is defined as δ with $\delta \in Z^{+},\delta ⩾4$. Commonly, when the number of extracted PIPs m is large enough (m=n), all points in T will be identified as PIPs, but if the parameter δ is considered, the upper limit of PIPs will be further restricted. The calculation steps of GRM-PIPs are as follow.*

***Step 1***
*: Put the first point $P_{1}\left(x_{1},y_{1}\right)$ and the last point $P_{n}\left(x_{n},y_{n}\right)$ in T as initial two PIPs into $L_{p}$.*

***Step 2***
*: Set a temporary PIP $P_{t}$, which can be any point in T, and calculate the vertical distance $VD_{t}$ between $P_{t}$ and the line $P_{1}P_{n}$. $P_{t}$ divides the sequence {$P_{1}$, ..., $P_{n}$} into two subsequences: {$P_{1}$, ..., $P_{t}$} and {$P_{t}$, ..., $P_{n}$}. If the length of any subsequence is less than δ, the current $P_{t}$ should not be considered, and a new point is set as $P_{t}$ to continue the calculation until a $P_{t}$ is found that can maximize $VD_{t}$ and satisfy that the length of all subsequences is greater than δ, then save this $P_{t}$ in $L_{p}$ as the third PIP.*

***Step 3***
*: The fourth PIP is the point that maximizes the vertical distance to its adjacent PIPs (which are either the first and the third, or the third and the second PIP) and controls the length of all segmented subsequences are greater than δ. It is also necessary to save the fourth PIP into the $L_{p}$.*

***Step 4***
*: For each new PIP, use the same recursive method as the fourth PIP, repeat Step 3 until the length of $L_{p}$ is equal to m.*


GRM-PIPs ensure a well-distribution of PIPs in the entire time series by adding a restriction on the interval length. A simple example in Figure 5 is shown to distinguish between the traditional PIPs and the GRM-PIPs proposed by us.
(2)$$T=\left\{0,1,2,10,9,10,9,9,6,4,3,1,5,3,10,10,8,9,10,11,9,6,3,0\right\}$$

In this example, we set a time series sample *T* in (7) with the length n=23. Figure 5 shows that the morphological structure of *T* is composed of two peaks and one trough. Seven PIPs were extracted from it. There is an apparent difference between the results of GRM-PIPs and PIPs, which are highlighted by red and green circles, respectively. Traditional PIPs are easy to fall into local optima because there is no interval constraint, and the selected PIPs do not contribute to the depiction of the overall structure. GRM-PIPs avoids this problem and accurately extracts PIPs that are more conducive to generalizing structural features.

In GRM-PIPs, affected by the length of interval δ, the number of extracted PIPs has an upper limit. In order to calculate the upper limit, we need to define the “quotient” first. Suppose there are two integers a and b, b≠0, there must be a pair of integers q and r satisfy a=q·b+r, and q can be called the quotient of a divided by b, abbreviated as $q=Q\left(a,b\right)$. In this way, the number of extracted PIPs can be calculated as follows:(3)$$2⩽m⩽Q\left(n,\delta \right)+2$$

It is obvious that the upper limit is closely related to δ. In our research, we set δ=4 because the subsequent feature extraction determines this value. We would explain the reason in detail in Section 3.2.

### 3.2. Feature Extraction

In this paper, we extract two features from time series, including point-level features $F_{P}$ and interval-level features $F_{I}$.

The point-level feature is straightforward, which is the coordinates of PIPs. We found that for different classes of time series, the distributions of PIPs in the two-dimensional space are also significantly different. Most importantly, these special distributions are consistent on the training set and the test set. Therefore, the point-level feature is distinctive and consistent, should be taken seriously. Some representative UCR datasets shown in Figure 6 confirm our views.

On the other hand, PIPs can generate excellent time series segmentation. Many datasets have no significant differences in the distribution of PIPs. At this time, the interval-level features need to be supplemented to help the classifier further distinguish samples of different categories. There are five interval-level features used by us:**Arithmetic mean**. The arithmetic mean (or simply mean) $\bar{x}$ of a sequence is the sum of all of the amplitudes divided by the length of the sequence *n*. This is a rough feature used to describe the average level of all data in the sequence. The calculation of the arithmetic mean follows Formula (4).
(4)$$\bar{x}=\frac{1}{n}\left(\sum_{i=1}^{n}x_{i}\right)=\frac{x_{1}+x_{2}+\cdot \cdot \cdot +x_{n}}{n}$$**Standard deviation**. In statistics, the standard deviation σ is a measure of the amount of variation or dispersion of a set of values. A low standard deviation indicates that the values tend to be close to the mean of the set, while a high standard deviation indicates that the values are spread out over a wider range. The standard deviation plays an important role in distinguishing frequently fluctuating series from stable changing series. The calculation of this feature is shown below.
(5)$$\sigma =\sqrt{\frac{1}{n}\sum_{i=1}^{n}{\left(x_{i}-\bar{x}\right)}^{2}}$$**Slope**. In mathematics, the slope or gradient of a line is a number that describes both the direction and the steepness of the line. Slope is calculated by finding the ratio of the “vertical change” to the “horizontal change” between (any) two distinct points on a line. We can also abstract any subsequence as a straight line connecting two adjacent PIPs and the trend can be judged by calculating the slope of the interval. For sequence $S=\left\{\left(x_{1},y_{1}\right),\ldots ,\left(x_{n},y_{n}\right)\right\}$, its slope can be calculated according to the following formula.
(6)$$m=\frac{\Delta y}{\Delta x}=\frac{y_{n}-y_{1}}{x_{n}-x_{1}}$$**Kurtosis**. In probability theory and statistics, kurtosis is a measure of the “tailedness” of the probability distribution of a real-valued random variable. The standard measure of a distribution’s kurtosis is a scaled version of the fourth moment of the distribution. Objectively speaking, kurtosis is not exactly the same as peakedness. Higher kurtosis means that the data has large deviations or extreme abnormal points, which deviate from the mean. However, in most cases, when the amplitude in a period of time in the time series is high, the corresponding kurtosis is high. In the calculation of kurtosis $G_{2}$ we use Standard unbiased estimator. It is worth noting that *n* represents the number of samples, and the formula needs to calculate $\left(n-3\right)$. As part of the denominator, it is required to be $\left(n-3\right)\neq 0$, which means that *n* must be a positive integer greater than 3. This is why we require the parameter δ to be equal to 4.
(7)$$\begin{array}{cc} G_{2} & =\frac{k_{4}}{k_{2}^{2}}=\frac{n^{2}\left[\left(n+1\right)m_{4}-3\left(n-1\right)m_{2}^{2}\right]}{\left(n-1\right)\left(n-2\right)\left(n-3\right)}\cdot \frac{{\left(n-1\right)}^{2}}{n^{2}m_{2}^{2}}\\ \; & =\frac{\left(n+1\right)n}{\left(n-1\right)\left(n-2\right)\left(n-3\right)}\cdot \frac{\sum_{i=1}^{n}{\left(x_{i}-\bar{x}\right)}^{4}}{k_{2}^{2}}-3\cdot \frac{{\left(n-1\right)}^{2}}{\left(n-2\right)\left(n-3\right)} \end{array}$$**Skewness**. In probability theory and statistics, skewness is a measure of the asymmetry of the probability distribution of a real-valued random variable about its mean. Skewness can be visually understood as the degree of inclination of the shape to the left or right. For example, in the two sequences shown in Figure 7, $S_{2}$ is almost obtained by mirror flipping of $S_{1}$, which is an indistinguishable situation for the mean, standard deviation, slope, and kurtosis. The use of skewness makes up for this deficiency. The calculation formula of skewness $G_{1}$ is similar to kurtosis, is a scaled version of the third central moment.
(8)$$\begin{array}{cc} G_{1} & =\frac{k_{3}}{k_{2}^{3/2}}=\frac{n^{2}}{\left(n-1\right)\left(n-2\right)}\cdot b_{1}=\frac{n^{2}}{\left(n-1\right)\left(n-2\right)}\cdot \frac{m_{3}}{\sigma^{3}}\\ \; & =\frac{n^{2}}{\left(n-1\right)\left(n-2\right)}\cdot \frac{\frac{1}{n}\sum_{i=1}^{n}{\left(x_{i}-\bar{x}\right)}^{3}}{{\left[\frac{1}{n-1}\sum_{i=1}^{n}{\left(x_{i}-\bar{x}\right)}^{2}\right]}^{3/2}} \end{array}$$

### 3.3. Classifer and the PFC Framework

In the TSC dataset, the data format is $D=\left\{data,label\right\}=\left\{\left(T_{1},\ldots ,T_{d}\right),\left(L_{1},\ldots ,L_{d}\right)\right\}$ with *d* time series and corresponding labels. We extract *m* PIPs through GRM-PIPs, and get m−1 intervals, thereby converting the original dataset into the corresponding feature set $F\left(D\right)=\left\{F_{P},F_{I}\right\}$. Subsequently, the training set in the $F\left(D\right)$ is input into the classifier and the test set is used for verification.

We realize that $F\left(D\right)$ is a high-level representation of raw data, essentially a combination of many features, and an explicit expression of morphological information. Therefore, we are more inclined to choose a classifier that is conducive to feature processing. In the PFC framework, we have selected three levels of classifiers, which are the decision tree as the basic estimator, the random forest with bagging idea, and the gradient boosting decision tree using boosting theory.

There are many ways to implement decision trees, such as ID3, C4.5, and CART. Under normal circumstances, the effect of CART is better than other methods, so we decided to implement CART. The reason for choosing RF and GBDT is that they are classifiers developed based on decision trees. RF conducts joint learning by constructing a large number of decision trees and integrates all classification results. RF equalizes the weights of all basic estimators, while GBDT gradually upgrades the weak classifiers to robust classifiers by iteratively changing the weights.

A schematic diagram of the PFC framework is shown in Figure 8. The innovation of our work is to propose GRM-PIPs, extract the combination of point-level and interval-level features, and use a suitable classifier to form a framework for TSC tasks. What we want to explore is the effect of the entire framework. Therefore, we did not make any special optimizations to the classifiers, and all the classifiers use traditional implementation methods. Further improvement of the classifier is our future work.

## 4. Performance Evaluation and Discussion

### 4.1. Experimental Design

The UCR archive has been widely used as a benchmark to evaluate TSC algorithms [8] (check details in http://www.timeseriesclassification.com, accessed on 1 May 2021). It currently contains 128 datasets, 15 of these are unequal length, 15 of there are missing values, and one (Fungi) has a single instance per class in the train files. Given this situation, in order to evaluate PFC, we select part of the UCR dataset. Since the two-category data is typically exclusive to each other, we divide the verification into two types, two-category and hybrid.

In the verification of two-category, we selected all the two-category datasets in UCR Archive and excluded the two with many missing values. Finally, 40 datasets were used for comparison experiments. Considering that PFC is a fast and straightforward classification method, it is unfair compared with some methods that use neural networks and consume substantial computing resources and time. Therefore, we exclude some deep learning algorithms for the benchmark model, such as ResNet and HIVE-COTE. The following five classification algorithms were selected for comparison, including the word extraction for time series classification (WEASEL), bag of symbolic-fourier approximation symbols (BOSS), time series forest (TSF), random interval spectral ensemble (RISE), and canonical time-series characteristics (Catch22). The results of these comparison algorithms have been officially recognized and released.

In the hybrid verification, we introduced some methods published recently as comparisons. These methods include extreme-SAX (E-SAX, 2020) [56], interval feature transformation (IFT, 2020) [57], and discriminative virtual sequences learning (DVSL, 2020) [58]. PFC is tested on the same datasets with these methods, including two-category datasets and multi-category datasets.

In addition, through the analysis of the experimental results, we would find answers to the following questions:What is the appropriate number of PIPs? The more always means the better?Does the number of PIPs have the same effect on different classifiers?

All experiments strictly follow UCR’s division of training set and test set. The classification accuracy is uniformly adopted as the metric. Some methods use classification errors and we convert them to accuracy. The number of time series correctly classified is defined as $n_{c}$, and the total number of time series of test set denoted by $n_{t}$. The calculation formula for classification accuracy (ACC) and error (ERR) is shown below.
(9)$$ACC=\frac{n_{c}}{n_{t}},ERR=1-ACC$$

Due to the randomness of RF and GBDT, the final experimental result is an average of 50 runs under the same parameters. At the same time, we do not do particular parameter optimization for DT, RF, and GBDT. DT uses the information gain to measure the quality of a split, and the nodes are expanded until all leaves are pure. There are 600 trees in RF, and the number of boosting stages to perform in GBDT is 600, too.

### 4.2. The Verification of Two-Category

The information of 40 two-category datasets in UCR Archive is listed in Table 2. Obviously, these datasets cover various situations such as short-sequence classification (Chinatown and ItalyPowerDemand), long-sequence classification (HandOutlines, HouseTwenty, and SemgHandGenderCh2), unbalanced training set and test set (ECGFiveDays and FreezerSmallTrain), and so on.

The classification accuracy of the five benchmark methods and PFC on these datasets is shown in Table 3. We found that not all datasets have public results on the five benchmark methods, and the results of two datasets (FordB and HandOutLines) are missing. These two datasets were excluded when calculating the number of times to obtain the best accuracy, and the experimental results of the remaining 38 datasets were considered.

The PFC framework achieved the best accuracy in 13 of 38 UCR datasets. What is interesting is that when DT and GBDT are used as classifiers, 6 times catch the best, which is less than 10 times when RF is used. Nevertheless, their performance has been better than RISE, TSF and Catch22.

This seems to be a counter-intuitive result. As the most complex classifier, GBDT has not achieved the best results. However, this situation can be explained. We noticed that there is a significant difference in the number of PIPs extracted by the GRM-PIPs algorithm when the best results are obtained (for detail see Appendix A). When DT and GBDT achieve their best results, the number of PIPs is almost the same, while RF requires more PIPs to achieve higher accuracy. This means that the upper limit of RF performance is the highest among the three classifiers. This may be caused by no parameter optimization. GBDT and DT usually rely on adjusting parameters to improve accuracy, while RF is not sensitive to parameters, and a large number of random decisions can effectively compensate for parameter defects.

We conduct an in-depth analysis of the experimental results shown in Table 3, which are divided into two aspects:**The impact of the length of the time series on accuracy**. We sort all the datasets according to their length, and the ones with a length less than 100 are classified as a group of $G_{1}$, which contains 11 datasets. $G_{2}$ has 11 datasets, the corresponding length is greater than 100 but less than 300. $G_{3}$ covers 15 datasets ranging in length from 300 to 1000. The remaining three datasets whose length exceeds 1000 are set as $G_{4}$. From $G_{1}$ to $G_{4}$, the number of times that PFC achieves the best accuracy is 3, 6, 4, and 0, respectively. The results show that PFC is good at distinguishing time series samples whose length ranges from 100 to 1000. For samples with a length less than 100, GRM-PIPs can only extract 27 PIPs at most and generate 26 intervals, which results in the feature dimension being much larger than the original sequence dimension, and the information redundancy makes the classifier unable to obtain robust decision rules. On the other hand, since we set up to extract only 30 PIPs in the experiment, the features of samples longer than 1000 may be incompletely extracted.**Does the imbalance of the training set and test set affect the accuracy of PFC**? As far as the current results are concerned, the training set and test set are not factors that affect accuracy.

### 4.3. The Hybrid Verification

In hybrid verification, we will compare with the TSC methods in three recently published papers. Since the datasets validated by these methods are different, we decided to compare them one by one and use the same datasets.

First, we test the performance of PFC and DVSL. Abhilash et al. [58] believed that the existing VSML methods employ fixed virtual sequences, which might not be optimal for the subsequent classification tasks. Therefore, they proposed DVSL to learn a set of discriminative virtual sequences that help separate time series samples in a feature space. Finally, this method was validated on 15 UCR datasets. The results of the comparative experiment are shown in Table 4.

The results show that PFC performed better on the same 15 UCR datasets and surpassed DVSL for the best accuracy in 12 of them. At the same time, we also notice that the accuracy of PFC is much lower than that of DVSL in datasets such as Beef. Figure 9 shows the distribution of PIPs in Beef. We can clearly find that only the distribution of Label=1 (represented by the red dots) is distinguishable, and the distributions of the other categories are highly similar. We believe that PFC can distinguish some samples with obvious distinguishing characteristics, but if these characteristics are highly similar in multiple types of samples, PFC will be invalid. Although this situation is accidental, PFC is based on morphological perception information, and it is difficult to process samples with small differences in morphology.

The second comparison method is IFT [57], which also uses PIPs. The difference is that IFT adopts information gain-based selection for interval features, which makes the whole method a special decision tree. Since both PFC and IFT perceive the importance of morphological features, this is a meaningful comparative experiment. IFT was validated on 22 UCR datasets, and we also tested on the same datasets. The comparison results are shown in Table 5.

On these datasets, the performance of PFC almost completely surpasses IFT. However, one exception to the results, was the huge difference in the accuracy of the PFC and IFT on a dataset called ShapeletSim. The samples in ShapeletSim present a form similar to high-frequency sinusoidal signals, which causes most of the PIPs to be located at the peaks and troughs. At this time, the distribution of PIPs can describe the boundary of the sample only, a rectangle in Figure 10. The crux of the problem is not just the abnormality of these distributions, we realize that they lack the necessary distinguishability. On this dataset, the performance of IFT is almost perfect. The reason may be that its feature selection is different from our work, and these unique features play an important role in classification.

Finally, we set our sights on E-SAX. One of the most popular dimensionality reduction techniques of time series data is the Symbolic Aggregate Approximation (SAX), which is inspired by algorithms from text mining and bioinformatics. E-SAX uses only the extreme points of each segment to represent the time series [56]. The essence of SAX is to reduce the dimensionality of time series, which is the same as PIPs. For these reasons, we chose E-SAX as the comparison method.

There are 45 UCR datasets used for comparison experiments, and all the results are listed in Table 6. It is important to point out that E-SAX originally used classification error ERR as the metric. In order to facilitate comparison, we convert the classification error ERR into classification accuracy ACC according to formula (9).

As shown in Table 6, the PFC achieves the most best ACC, and best performance in 34 out of 45 datasets.These datasets are divided into 17 two-category datasets and 28 multi-category datasets. PFC has achieved significant advantages in 13 two-category datasets and 21 multi-category datasets. Although PFC is still at a disadvantage in some datasets, we found that the results obtained by PFC are very close to E-SAX, which is based on the premise that we have not optimized any parameters and model structure. We believe that PFC still has the possibility of improvement.

This comparison experiment and the previous two-category verification have a very small gap in the number of datasets used. It is equivalent to removing part of the two-category datasets and introducing a large number of multi-category datasets based on the latter. However, the number of times that the PFC using RF as a classifier achieves the best accuracy has greatly increased, far exceeding the cases of DT and GBDT. RF can rely on a large number of decision trees to satisfy multi-classification tasks, and this advantage has been demonstrated.

### 4.4. Discussion on the Number of PIPs

This is a meaningful discussion, because most of the current papers ignore this problem. No matter what operations will be performed later, we usually extract *m* PIPs from the original time series at first. There are two questions that need to be answered at this time:What is the appropriate number of *m*? The more always means the better?Does the number of PIPs have the same effect on different classifiers?

The second question is relatively easy to answer. The data listed in Appendix A. gives us the answer: the same *m* has different effects on different classifiers. RF and GBDT always require a large number of PIPs to achieve high accuracy, but DT is not so demanding. RF and GBDT as ensemble methods must be suitable for more features, but on some simple datasets, DT can outperform them with a few PIPs.

In fact, the most difficult thing is to answer the first question. As shown in Figure 11, with the length of the dataset as the horizontal axis, we obtain the distribution of PIPs when the best accuracy is achieved on the corresponding dataset. The three distributions are similar, but for RF and GBDT, the appropriate number of PIPs is greater than DT.

On the other hand, on the same dataset, the larger *m* does not mean the better. Through a large number of experimental records, we found that there is no specific rule. For some time series with quite different morphological structures, a small amount of PIPs is enough to highlight their differences. Conversely, more PIPs may cause information redundancy and confusion. When the morphological structure of the time series is complex, the situation is completely opposite, and more PIPs are needed to describe the characteristics of the sample.

## 5. Conclusions

The introduction of morphological structure features is an important improvement to the time series classification. Based on the way of human visual cognition, many studies have pointed out that the shape of time series can be described by a sequence of important turning points. Inspired by these studies, we proposed GRM-PIPs, which control the length of the interval. Then we used PIPs to segment the time series, and extracted the feature combination of interval-level and point-level. The introduction of three classifiers, DT, RF, and GBDT, completes the perceptual feature-based framework. Finally, we compared five benchmark methods and three recently published methods on a large number of UCR datasets. The experimental results show that our work has excellent performance on the TSC task. In addition, we demonstrated the threshold of the interval length and discussed the influence of the number of PIPs, which made up for the deficiency in these aspects.

In future work, we plan to add more different types of classifiers and optimize these classifiers. At the same time, further improvement of feature extraction is considered.

## Figures and Tables

**Figure 1 entropy-23-01059-f001:**
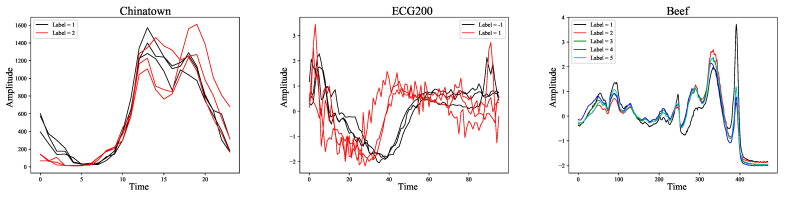
Three representative datasets from the UCR Time Series Classification Archive. Due to the large size of the original dataset, only some samples are shown as examples.

**Figure 2 entropy-23-01059-f002:**
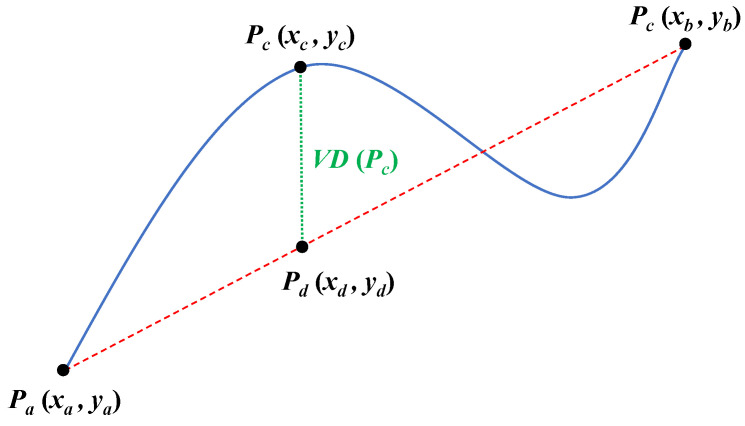
The schematic diagram of the vertical distance $VD\left(P_{c}\right)$.

**Figure 3 entropy-23-01059-f003:**
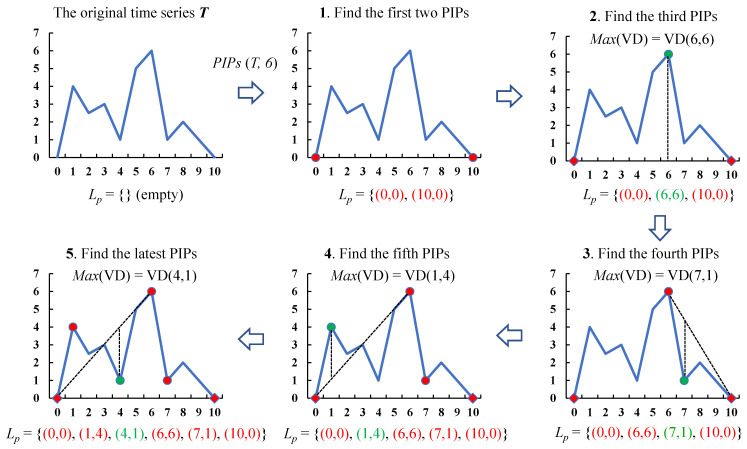
The process of finding 6 PIPs from *T*.

**Figure 4 entropy-23-01059-f004:**
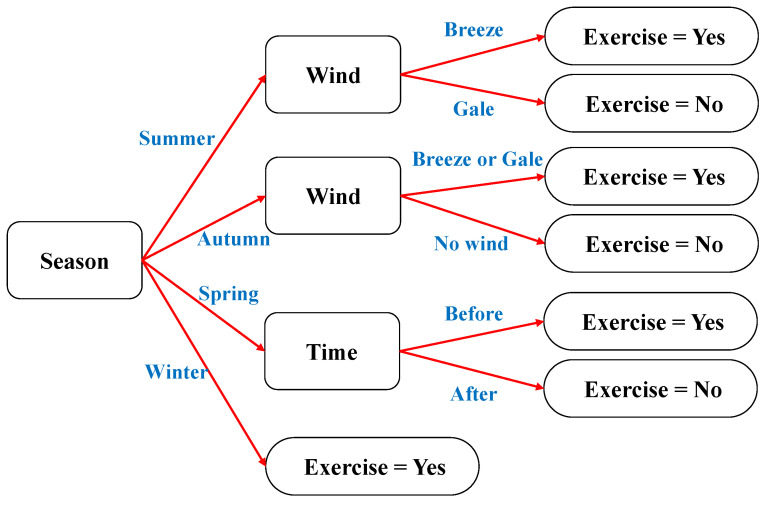
A decision tree constructed based on the example.

**Figure 5 entropy-23-01059-f005:**
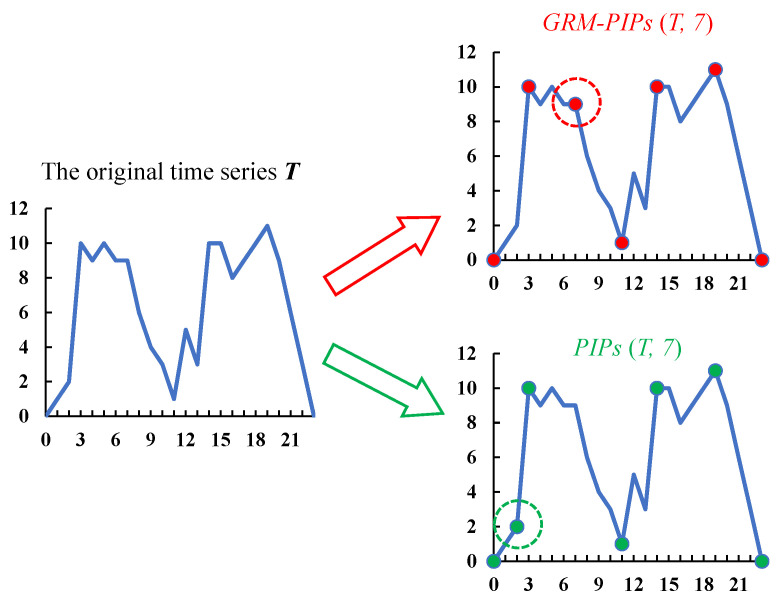
GRM-PIPs and the traditional PIPs algorithms were used to extract PIPs from time series sample *T*.

**Figure 6 entropy-23-01059-f006:**
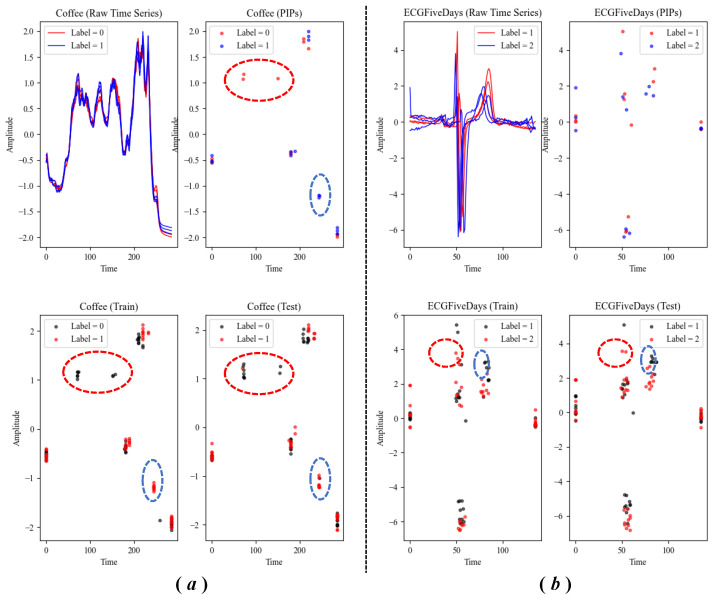
The distributions of PIPs in two UCR datasets, which are Coffee (**a**) and ECGFiveDays (**b**). The figures above show that the PIPs extracted from the original sample are discriminative, while the figures below show that the distribution of PIPs is consistent on the training set and the test set.

**Figure 7 entropy-23-01059-f007:**
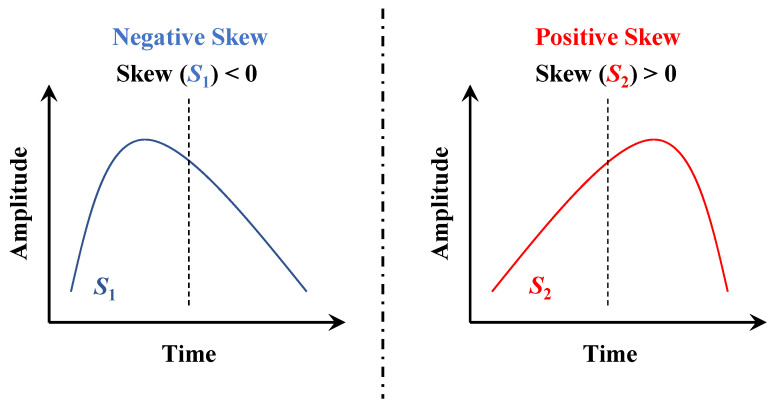
An instance that cannot be distinguished by features such as mean and standard deviation. The sequence $S_{1}$ on the left is flipped to get the sequence $S_{2}$ on the right.

**Figure 8 entropy-23-01059-f008:**
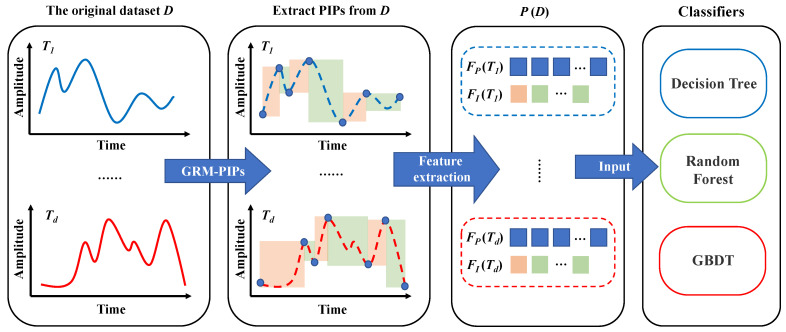
The schematic diagram of the PFC framework.

**Figure 9 entropy-23-01059-f009:**
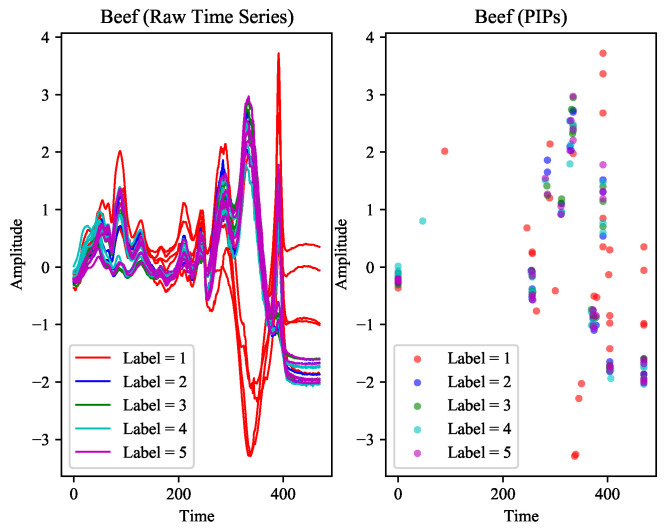
The original time series and the distribution of PIPs in Beef.

**Figure 10 entropy-23-01059-f010:**
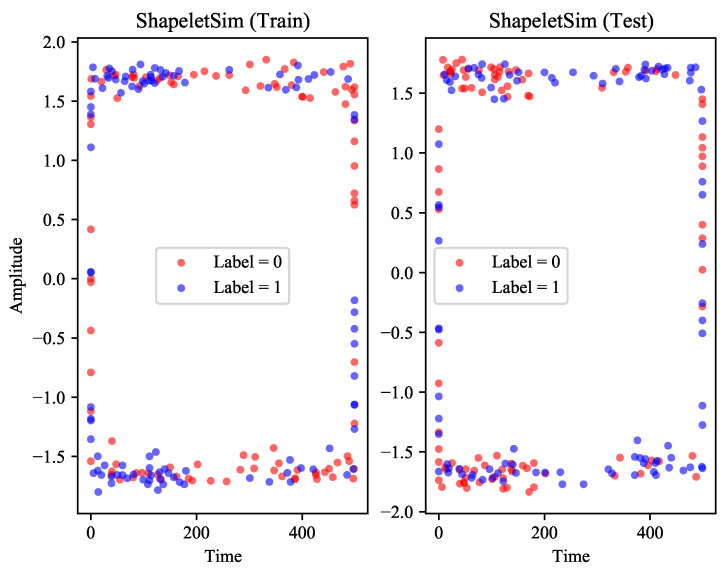
The distribution of PIPs in ShapeletSim. The distribution on the training set is on the left, and the right is the distribution on the test set.

**Figure 11 entropy-23-01059-f011:**
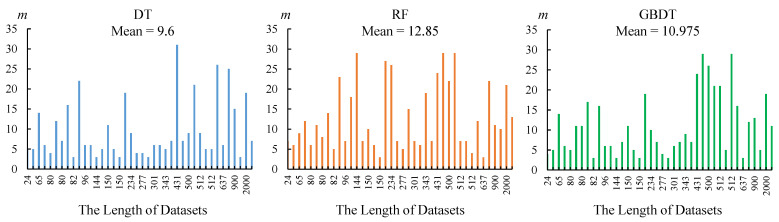
The distribution of the number of PIPs in different classifers.

**Table 1 entropy-23-01059-t001:** The sample data of DT.

Season	Time	Wind	Exercise
Spring	Before 8:00 a.m.	Breeze	Yes
Winter	Before 8:00 a.m.	No wind	Yes
Autumn	After 8:00 a.m.	Breeze	Yes
Winter	Before 8:00 a.m.	No wind	Yes
Summer	Before 8:00 a.m.	Breeze	Yes
Winter	After 8:00 a.m.	Breeze	Yes
Winter	Before 8:00 a.m.	Gale	Yes
Winter	Before 8:00 a.m.	No wind	Yes
Spring	After 8:00 a.m.	No wind	No
Summer	After 8:00 a.m.	Gale	No
Summer	Before 8:00 a.m.	Gale	No
Autumn	After 8:00 AM	Breeze	No

**Table 2 entropy-23-01059-t002:** Summary of 40 two-category datasets in UCR Archive.

No.	Name	Type	Train	Test	Length
1	BeetleFly	Image	20	20	512
2	BirdChicken	Image	20	20	512
3	Chinatown	Traffic	20	343	24
4	Coffee	Spectro	28	28	286
5	Computers	Device	250	250	720
6	DistalPhalanxOutlineCorrect	Image	600	276	80
7	Earthquakes	Sensor	322	139	512
8	ECG200	ECG	100	100	96
9	ECGFiveDays	ECG	23	861	136
10	FordA	Sensor	3601	1320	500
11	FordB	Sensor	3636	810	500
12	FreezerRegularTrain	Sensor	150	2850	301
13	FreezerSmallTrain	Sensor	28	2850	301
14	GunPoint	Motion	50	150	150
15	GunPointAgeSpan	Motion	135	316	150
16	GunPointMaleVersusFemale	Motion	135	316	150
17	GunPointOldVersusYoung	Motion	136	315	150
18	Ham	Spectro	109	105	431
19	HandOutlines	Image	1000	370	2709
20	Herring	Image	64	64	512
21	HouseTwenty	Device	40	119	2000
22	ItalyPowerDemand	Sensor	67	1029	24
23	Lightning2	Sensor	60	61	637
24	MiddlePhalanxOutlineCorrect	Image	600	291	80
25	MoteStrain	Sensor	20	1252	84
26	PhalangesOutlinesCorrect	Image	1800	858	80
27	PowerCons	Power	180	180	144
28	ProximalPhalanxOutlineCorrect	Image	600	291	80
29	SemgHandGenderCh2	Spectrum	300	600	1500
30	ShapeletSim	Simulated	20	180	500
31	SonyAIBORobotSurface1	Sensor	20	601	70
32	SonyAIBORobotSurface2	Sensor	27	953	65
33	Strawberry	Spectro	613	370	235
34	ToeSegmentation1	Motion	40	228	277
35	ToeSegmentation2	Motion	36	130	343
36	TwoLeadECG	ECG	23	1139	82
37	Wafer	Sensor	1000	6164	152
38	Wine	Spectro	57	54	234
39	WormsTwoClass	Motion	181	77	900
40	Yoga	Image	300	3000	426

**Table 3 entropy-23-01059-t003:** Classification accuracy of PFC and five benchmarks on 40 two-category UCR datasets.

							PFC	
No.	WEASEL	BOSS	RISE	TSF	Catch22	PFC-DT	PFC-RF	PFC-GBDT
1	0.8867	0.9433	0.8717	0.8333	0.8400	0.9000	**0.9500**	0.9000
2	0.8650	**0.9833**	0.8683	0.8150	0.8933	0.9500	0.9000	0.9500
3	0.9573	0.8771	0.8885	0.9530	0.9345	**0.9795**	**0.9795**	0.9767
4	0.9893	0.9857	0.9845	0.9869	0.9798	**1.0000**	**1.0000**	**1.0000**
5	0.7785	**0.8005**	0.7789	0.6488	0.7803	0.7000	0.7400	0.7640
6	**0.8192**	0.8117	0.8112	0.8058	0.8121	0.7500	0.7899	0.7753
7	0.7475	0.7460	0.7482	0.7475	0.7388	0.7913	**0.8058**	0.7986
8	0.8590	**0.8783**	0.8510	0.8600	0.7887	0.8100	0.8600	0.8500
9	0.9935	0.9923	0.9729	0.9520	0.8159	**0.9988**	0.9501	0.9954
10	**0.9687**	0.9214	0.9400	0.8158	0.9092	0.7136	0.8530	0.8734
11	lack	lack	lack	lack	lack	0.6432	0.7025	0.7271
12	0.9906	0.9881	0.9523	0.9971	**0.9982**	0.9782	0.9621	0.9775
13	0.9006	**0.9616**	0.8787	0.9614	0.9598	0.9281	0.9081	0.9421
14	0.9931	**0.9964**	0.9809	0.9553	0.9431	0.9533	0.9933	0.9533
15	0.9813	**0.9949**	0.9863	0.9777	0.9439	0.9589	0.9936	0.9873
16	0.9939	0.9996	0.9911	0.9960	0.9935	0.9810	**1.0000**	0.9810
17	0.9860	0.9992	0.9998	**1.0000**	0.9642	**1.0000**	**1.0000**	**1.0000**
18	0.8213	**0.8375**	0.8197	0.7994	0.6940	0.6667	0.7714	0.7486
19	lack	lack	lack	lack	lack	0.8865	0.9216	0.9351
20	0.6021	0.5958	0.5984	0.6042	0.5557	**0.6875**	0.6563	**0.6875**
21	0.8106	**0.9560**	0.9297	0.8378	0.9462	0.8740	0.9243	0.8740
22	0.9468	0.8709	0.9445	**0.9595**	0.8775	0.9417	0.9485	0.9105
23	0.6273	0.8191	0.6820	0.7645	0.7448	0.7541	**0.8197**	**0.8197**
24	0.8283	0.8095	0.8055	0.7995	0.7727	0.7423	**0.8316**	0.8178
25	**0.9048**	0.8442	0.8780	0.8555	0.8485	0.7764	0.8594	0.7572
26	**0.8217**	0.8174	0.8125	0.8057	0.7919	0.7145	0.7995	0.8007
27	0.9194	0.8900	0.9580	**0.9931**	0.8863	0.9333	0.9556	0.9500
28	0.8763	0.8655	0.8737	0.8489	0.8337	0.8419	0.8797	**0.8965**
29	0.7814	0.8877	0.8700	**0.9474**	0.8706	0.8250	0.8867	0.8867
30	0.9974	**1.0000**	0.7676	0.5137	0.9937	0.5667	0.5944	0.5667
31	**0.9093**	0.8977	0.8670	0.8637	0.8834	0.8469	0.7804	0.8469
32	**0.9353**	0.8794	0.9125	0.8743	0.9023	0.8001	0.7827	0.7901
33	0.9786	0.9705	0.9730	0.9675	0.9229	0.9108	**0.9824**	0.9622
34	**0.9430**	0.9249	0.8804	0.6671	0.8127	0.7675	0.8915	0.8333
35	0.9285	**0.9615**	0.9118	0.8026	0.8351	0.7692	0.8308	0.7462
36	**0.9975**	0.9847	0.9107	0.8706	0.8539	0.9543	0.9719	0.9543
37	0.9999	0.9989	0.9954	0.9966	0.9973	**1.0000**	**1.0000**	**1.0000**
38	**0.9302**	0.8926	0.8710	0.8623	0.7000	0.7037	0.7963	0.7222
39	0.8004	**0.8078**	0.7853	0.6935	0.7922	0.7143	0.7403	0.7800
40	0.8924	**0.9102**	0.8372	0.8658	0.8038	0.7497	0.8200	0.8097
Best ACC	9	12	0	4	1		**13** (6/10/6)	

**Table 4 entropy-23-01059-t004:** Comparison of PFC and DVSL on 15 UCR datasets.

							PFC	
Datasets	Train	Test	Class	Length	DVSL	PFC-DT	PFC-RF	PFC-GBDT
ArrowHead	36	175	3	251	0.7200	0.6114	**0.7257**	**0.7257**
Beef	30	30	5	470	**0.9000**	0.7000	0.7333	0.6667
Car	60	60	4	577	**0.8350**	0.7167	0.7167	0.7500
ChlConcent	467	3840	3	166	**0.7743**	0.5951	0.6497	0.6466
Coffee	28	28	2	286	**1.0000**	**1.0000**	**1.0000**	**1.0000**
ECG200	100	100	2	96	0.8350	0.8100	**0.8600**	0.8500
ECGFiveDays	23	861	2	136	0.9735	**0.9988**	0.9501	0.9954
Herring	64	64	2	512	0.6563	**0.6875**	0.6563	**0.6875**
InsectWingb	220	1980	11	256	0.5819	0.4217	**0.5927**	0.4788
Meat	60	60	3	448	0.9883	**1.0000**	0.9667	0.9667
MPhaOLAge	400	154	3	80	0.5818	0.5779	**0.6558**	0.5974
OliveOil	30	30	4	570	0.8467	**0.9667**	0.9333	0.9000
SonyAIBR1	20	601	2	70	0.7616	**0.8469**	0.7804	**0.8469**
TwoLeadECG	23	1139	2	82	0.9160	0.9543	**0.9719**	0.9543
Wine	57	54	2	234	0.6500	0.7037	**0.7963**	0.7222
Best ACC					4		**12** (6/7/4)	

**Table 5 entropy-23-01059-t005:** Comparison of PFC and IFT on 20 UCR datasets (exclude two datasets with missing values).

							PFC	
Datesets	Train	Test	Length	Class	IFT	PFC-DT	PFC-RF	PFC-GBDT
BirdChicken	20	20	512	2	0.9000	**0.9500**	0.9000	0.9500
FreezerRegularTrain	150	2850	301	2	0.9035	**0.9782**	0.9621	0.9775
ShapeletSim	20	180	500	2	**0.9944**	0.5667	0.5944	0.5667
ToeSegmentation1	40	228	277	2	0.8816	0.7675	**0.8915**	0.8333
Worms	181	77	900	5	0.6623	0.5974	0.6623	**0.6883**
Rock	20	50	2844	4	0.6200	0.7000	**0.7600**	0.7000
Meat	60	60	448	3	0.9500	**1.0000**	0.9667	0.9667
Beef	30	30	470	5	0.6667	0.7000	**0.7333**	0.6667
InlineSkate	100	550	1882	7	0.3582	0.2800	**0.3818**	0.3327
Coffee	28	28	286	2	0.9643	**1.0000**	**1.0000**	**1.0000**
ECGFiveDays	23	861	136	2	0.8281	**0.9988**	0.9501	0.9954
Ham	109	105	431	2	0.6381	0.6667	**0.7714**	0.7486
Herring	64	64	512	2	0.6719	**0.6875**	0.6563	**0.6875**
PowerCons	180	180	144	2	0.9333	0.9333	**0.9556**	0.9500
Wine	57	54	234	2	0.7407	0.7037	**0.7963**	0.7222
Yoga	300	3000	426	2	0.7767	0.7497	**0.8200**	0.8097
FaceFour	24	88	350	4	0.6477	**0.7045**	0.6136	0.6591
OliveOil	30	30	570	4	0.7667	**0.9667**	0.9333	0.9000
Fish	175	175	463	7	0.8114	0.7600	**0.8971**	0.8500
Plane	105	105	144	7	**1.0000**	0.9429	**1.0000**	0.9905
Best ACC					2		**19** (8/9/4)	

**Table 6 entropy-23-01059-t006:** Comparison of PFC and E-SAX on 45 UCR datasets. All results are converted to accuracy uniformly.

							PFC	
Datesets	Train	Test	Class	Length	E-SAX	PFC-DT	PFC-RF	PFC-GBDT
SyntheticControl	300	300	6	60	**0.9970**	0.8400	0.9767	0.9500
GunPoint	50	150	2	150	0.8600	0.9533	**0.9933**	0.9533
CBF	30	900	3	128	0.9190	**0.9522**	0.9111	0.9056
FaceAll	560	1690	14	131	0.7250	0.6817	**0.8302**	0.8012
OSULeaf	200	242	6	427	0.5160	0.5248	**0.6612**	0.6157
SwedishLeaf	500	625	15	128	0.7520	0.7296	**0.8752**	0.8112
Trace	100	100	4	275	0.6800	**1.0000**	**1.0000**	**1.0000**
FaceFour	24	88	4	350	**0.7840**	0.7045	0.6136	0.6591
Lightning2	60	61	2	637	**0.8360**	0.7541	0.8197	0.8197
Lightning7	70	73	7	319	0.6020	0.6986	**0.7945**	0.7397
ECG200	100	100	2	96	**0.8800**	0.8100	0.8600	0.8500
Adiac	390	391	37	176	0.1460	0.4731	**0.6419**	0.5242
Yoga	300	3000	2	426	**0.8210**	0.7497	0.8200	0.8097
Fish	175	175	7	463	0.7540	0.7600	**0.8971**	0.8500
Plane	105	105	7	144	0.9710	0.9429	**1.0000**	0.9905
Car	60	60	4	577	0.7330	0.7167	0.7167	**0.7500**
Beef	30	30	5	470	0.6330	0.7000	**0.7333**	0.6667
Coffee	28	28	2	286	0.7140	**1.0000**	**1.0000**	**1.0000**
OliveOil	30	30	4	570	0.1670	**0.9667**	0.9333	0.9000
CinCECGTorso	40	1380	4	1639	0.9270	0.8800	**0.9415**	0.9223
ChlorineConcentration	467	3840	3	166	0.4920	0.5951	**0.6497**	0.6466
DiatomSizeReduction	16	306	4	345	**0.9120**	0.9052	0.9020	0.8889
ECGFiveDays	23	861	2	136	0.7650	**0.9988**	0.9501	0.9954
FacesUCR	200	2050	14	131	**0.7940**	0.5937	0.7405	0.6615
Haptics	155	308	5	1092	0.3380	0.3701	**0.5097**	0.4805
InlineSkate	100	550	7	1882	0.3300	0.2800	**0.3818**	0.3327
ItalyPowerDemand	67	1029	2	24	0.8880	0.9417	**0.9485**	0.9105
MedicalImages	381	760	10	99	0.6420	0.6618	**0.7421**	0.6868
MoteStrain	20	1252	2	84	0.8070	0.7764	**0.8594**	0.7572
SonyAIBORobotSurface1	20	601	2	70	0.6940	**0.8469**	0.7804	**0.8469**
SonyAIBORobotSurface2	27	953	2	65	**0.8540**	0.8001	0.7827	0.7901
Symbols	25	995	6	398	0.8970	0.8040	**0.9317**	0.8523
TwoLeadECG	23	1139	2	82	0.7220	0.9543	**0.9719**	0.9543
InsectWingbeatSound	220	1980	11	256	0.5470	0.4217	**0.5927**	0.4788
ArrowHead	36	175	3	251	**0.7770**	0.6114	0.7257	0.7257
BeetleFly	20	20	2	512	0.7500	0.9000	**0.9500**	0.9000
BirdChicken	20	20	2	512	0.6500	**0.9500**	0.9000	**0.9500**
Herring	64	64	2	512	0.5940	**0.6875**	0.6563	**0.6875**
ProximalPhalanxTW	400	205	6	80	0.6380	0.7366	**0.8195**	0.7951
ToeSegmentation1	40	228	2	277	0.6450	0.7675	**0.8915**	0.8333
ToeSegmentation2	36	130	2	343	0.8080	0.7692	**0.8308**	0.7462
DistalPhalanxOutlineAgeGroup	400	139	3	80	0.7500	0.7338	**0.7626**	0.7482
DistalPhalanxOutlineCorrect	600	276	2	80	0.6020	0.7500	**0.7899**	0.7753
DistalPhalanxTW	400	139	6	80	**0.7280**	0.6978	0.6906	0.6763
WordSynonyms	267	638	25	270	**0.6290**	0.4232	0.5799	0.4828
Best ACC					11		**34** (8/27/6)	

## Data Availability

The UCR dataset comes from https://www.cs.ucr.edu/~eamonn/time_series_data_2018/ (accessed on 1 May 2021). The complete data package can be downloaded from https://www.cs.ucr.edu/~eamonn/time_series_data_2018/UCRArchive_2018.zip (accessed on 1 May 2021). The briefing documents of the UCR dataset can be downloaded here (https://www.cs.ucr.edu/~eamonn/time_series_data_2018/BriefingDocument2018.pdf and https://www.cs.ucr.edu/~eamonn/time_series_data_2018/BriefingDocument2018.pptx) (accessed on 1 May 2021). More information about the UCR dataset such as baseline and comparison can be found in http://www.timeseriesclassification.com (accessed on 1 May 2021).

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
