# Peer review of "PFC: A Novel Perceptual Features-Based Framework for Time Series Classification"

_entropy, 2021, doi:10.3390/e23081059_

Round 1
Reviewer 1 Report
The description of some statistical tools, used in the article, is too detailed and should be shortened. In particular, the description of decision tree (and the example) in section 2,2 and the illustration of feature extraction in section 3.2 could be summarized (in my opinion, it is useless to provide the definition and formulas for mean, standard deviation, ...kurtosis...these are just included in any standard textbook)
Reviewer 2 Report
This manuscript presents a new method (PFC) for time series classification (TSC). Inspired by the way humans process information, the method is based on a perceptual features-based framework. The authors propose to obtained Perception Important Points (PIP) to characterize the time series, in order to reduce the dimensionality. Although this concept is not new (as the author address), the authors improve the PIP concept by introducing GRM-PIPs (PIPs separated by at least a threshold distance), in other to ensure a appropriate distribution of PIPs along the time series.
Once the PIPs are termined, two types of features are extracted from them, point-level and interval-level. These features are used as the inputs of three different classifiers, and the results are compared to other state-of-the-art techniques, using for that purpose a well-known time series data set (UCR Archive). In the majority of cases, the results of the present technique outperform other methods, which are often more complicated and more expense computationally than PFC.
In my opinion, this manuscript presents an excellent work in the field. The paper is well structured and written, the references are well selected and updated, the choice of other techniques to compare is correct, and more important, the results sound and the conclussions are well supported by them.
I considet that the only open question (although already discussed in the paper) is a possible authomatic non-supervised determination of the number m of PIPs required for achieving optimal results (or, equivalently, of the minimum distance delta between consecutive PIPs). I understand this is a critical question, that deserves further investigation (not in this work). As as suggestion, I would consider to use Fourier information of the corresponding time series to determine the frequency where the power spectrum reaches its maximum value, and use the inverse of such frequency as an initial estimation of delta.
In conclussion, I consider this manuscript as suitable for publication in Entropy, and I only suggest a careful English revision to eliminate some typos in the text.
Round 2
Reviewer 1 Report
accept in the present form
Reviewer 2 Report
I consider the manuscript suitable for publicación in Entropy after your minor chances.